# Can We Use Simple Radiographic Measurements to Predict Need for Intervention in Neonatal Pneumothorax?

**DOI:** 10.3390/children13010041

**Published:** 2025-12-27

**Authors:** Kati N. Baillie, Rohit Misra, Pauravi Vasavada, Moira Crowley, Monika Bhola, Rita M. Ryan

**Affiliations:** 1Rainbow Babies & Children’s Hospital, Case Western Reserve University, Cleveland, OH 44106, USA; katibaillie@gmail.com (K.N.B.); rohit.misra@uhhospitals.org (R.M.); moira.crowley@uhhospitals.org (M.C.); monika.bhola@uhhospitals.org (M.B.); 2Department of Pediatrics and Radiology, Case Western Reserve University, Cleveland, OH 44106, USA; pauravi.vasavada@uhhospitals.org

**Keywords:** thoracentesis, thoracostomy, chest tube, respiratory distress syndrome

## Abstract

**Background:** Pneumothorax (PTX) develops in 1–2% of neonates, leading to significant morbidity and mortality and requiring providers to be comfortable with management. Our objective was to evaluate whether radiographic measurements of PTX size can be used to predict the need for procedural intervention in neonates in order to help guide the need for the availability of specific personnel. **Methods:** With the help of a data analyst, 62 patients diagnosed with neonatal PTX between March 2016 and October 2024 were identified. Most babies (46) were born in 2023–2024 when our new electronic health record could more easily identify these infants. PTX size was evaluated using radiographs by calculating the ratio of the widest transverse measurement of the PTX on both anteroposterior (AP) and, when available, lateral decubitus (DECUB) divided by the widest transverse measurement of the hemithorax above the diaphragm. Clinical data were collected, and statistical analysis was performed using need for intervention (thoracentesis (TC), chest tube (CT), or both). **Results:** We found that a larger PTX size ratio, measured in the AP (*p* < 0.0001) or DECUB view (*p* < 0.008), was highly associated with need for intervention in this cohort of infants with PTX. Only 33% of PTXs required intervention. Also, 13/14 (93%) cases who underwent TC ultimately required a CT. PTX was more prevalent in males in general, but sex was not associated with needing intervention. The average gestational age (GA) of the cohort was 36 5/7 weeks, with only 12% being < 34 weeks GA. Univariate analysis indicated that lower GA and birth weight were risk factors for intervention. There was a trend (*p* = 0.075, by Fisher’s exact test) suggesting that infants with both respiratory distress syndrome (RDS) and PTX may be more likely (60%) to require intervention (no RDS, 29% intervention). Finally, a receiver operator characteristic curve was derived from the AP ratio based on the yes/no intervention which resulted in an area under the curve statistic of 0.902 and the optimal AP ratio cutoff of 0.184. **Conclusions:** The ratio of the transverse measurement of the PTX/hemithorax size from radiographs was highly predictive for need for intervention in a cohort of primarily term infants with PTX. Smaller and lower GA infants were at a higher risk for requiring procedural intervention. Nearly all infants who had TC also needed a CT. These findings could inform clinical strategies for managing neonatal PTXs, especially in identifying appropriate needed personnel availability if a TC occurs.

## 1. Introduction

Pneumothorax (PTX) is defined as the accumulation of air within the potential space between the visceral and parietal pleura, sometimes leading to lung collapse. It may occur spontaneously, in the absence of trauma or other external factors, or from complications related to mechanical ventilation, infection, trauma, or certain disease processes. Neonatal pneumothorax occurs in approximately 1–2% of neonates admitted to the neonatal intensive care unit (NICU), with higher rates in premature infants and those weighing < 1500 g [1,2]. Mortality is increased in neonates diagnosed with pneumothorax [3]. Pneumothorax in neonates is diagnosed via chest transillumination, chest radiography, and less commonly, ultrasonography [4,5].

While many neonatal pneumothoraces can be managed conservatively, symptomatic cases may require needle thoracentesis (TC) or chest tube (CT) placement [6]. In practice, small, asymptomatic pneumothoraces can be managed conservatively while larger, symptomatic pneumothoraces are managed procedurally [7,8,9]. However, criteria reliably predicting the need for procedural intervention based on size are limited. Data in pediatric and adult patients have established measurement cutoffs based on x-ray and CT measurements respectively to predict the need for procedural evacuation [9,10]. Similar data in the neonatal population is lacking.

Our objective was to evaluate whether radiographic two-dimensional measurements of pneumothorax size could be used to predict the need for procedural intervention in neonates. An easily replicable and simple radiographic ratio could potentially serve as a predictive tool to determine the need for intervention, helping clinicians make timely and accurate decisions in the NICU setting, including appropriate future availability of personnel with specific skills.

## 2. Materials and Methods

Hospital Information Technology personnel queried the electronic medical record (EMR) to identify a convenience sample of neonates diagnosed with pneumothorax between March 2016 and October 2024 in our single, large (82-bed), level IV NICU. The EMR search included International Classification of Disease (ICD) diagnostic codes for neonatal pneumothorax (e.g., “pneumothorax of newborn,” “pneumothorax originating in the neonatal period”) and Current Procedural Terminology (CPT) codes associated with thoracentesis and chest tube placement. A total of 71 patient charts were identified. We had changed our EMR on 30 September 2023 and so 46 of the 71 patients identified were from when the new EMR was in place; ascertainment using our former EMR proved more difficult. Inclusion criteria consisted of neonates admitted to the Rainbow Babies and Children’s NICU who had a pneumothorax sometime during admission. Exclusion criteria included neonates without pneumothorax or those with a confounding diagnosis, such as congenital diaphragmatic hernia, tracheoesophageal fistula, or congenital heart disease requiring surgery during newborn admission.

The primary outcome of the study was to determine if radiographic pneumothorax size ratio (PTX/hemidiaphragm) could predict the need for any procedural intervention. Secondary outcomes were to assess if clinical and demographic factors such as gestational age, birth weight, and associated diagnoses such as respiratory distress syndrome (RDS) could influence the likelihood of needing intervention.

To determine our sample size, we performed a sample size calculation. Pneumothorax size ratio, a continuous variable, was compared between two groups: those requiring intervention, and those requiring no intervention. Using a power of 80%, an alpha of <0.05 with a standard deviation of 0.128, and an assumption that approximately 30% of our PTX would need intervention, our estimated appropriate sample size included 63 total neonates; 19 of these required intervention and 44 required no intervention, to not miss a difference of 0.1 between the two groups. The standard deviation estimate of 0.128 was obtained by averaging five random neonatal pneumothorax measurements due to limited availability of any standard deviation data in the existing literature.

A retrospective chart review was conducted of the patients identified from the EMR query. Clinical data were extracted. Birth weight was obtained from the chart as the recorded birth weight in grams. Gestational age was calculated by recorded gestational age at birth, based on best obstetrical dating. In the absence of obstetrical dating, our routine practice is to perform a Ballard exam. Radiographs were reviewed using the Picture Archiving and Communication System (PACS). The size of the pneumothorax was assessed by measuring the widest transverse diameter of the pneumothorax on anteroposterior (AP) and, when available, lateral decubitus (DECUB) chest radiographs. This measurement was then divided by the widest transverse diameter of the hemithorax above the diaphragm to obtain a ratio (Figure 1). This measurement was modeled after similar measurements of pneumothorax radiographs in Ozer et al. 2013 [11]. A single investigator performed all radiographic measurements, so inter-rater reliability statistics were not assessed. The investigator was blinded to outcome.

Statistical analysis was performed to evaluate the association between pneumothorax size and the need for intervention (thoracentesis, chest tube placement, or both). To match what tended to be clinically available most consistently, most of the analysis and conclusions were done using the AP films and their subsequent ratios. Basic univariate analysis was done to compare the characteristics of the PTX-intervention and PTX-no intervention using either chi-square, Fisher’s exact test for categorical variables and *t*-test, or Wilcoxon rank sum test for continuous variables, as appropriate. The AP ratio and DECUB ratio were both examined for their association with intervention or no intervention using the Wilcoxon rank sum test to compare a non-normally distributed continuous variable between two groups. Descriptive data were examined to demonstrate a lower and upper cutoff at which no or all neonates received intervention (Figure 2). A receiver operator characteristic curve was derived from the ratio based on the yes/no intervention (Figure 3). Finally, to further test the utility of the anterior–posterior ratio (APr), we performed a backward stepwise multiple logistic regression analysis using a model that included the AP ratio, gestational age, use of CPAP in the delivery room (DR-CPAP), 5-min Apgar score, sex, and surfactant usage before the PTX was noted. With only 62 subjects, we limited the analysis to six variables in the model and chose the ones we thought would be most important.

## 3. Results

Of the initial 71 charts identified, eight charts were excluded from analysis due to insufficient patient information in the charts and one was excluded for an anterior pneumothorax that could not be measured using our criteria. The remaining 62 patient charts and their corresponding radiographs were used for analysis.

Pneumothorax was more prevalent in males, but sex was not associated with needing intervention. Average gestational age (GA) of our population was 36 5/7 weeks, with only 7/62 (11%) being < 34 weeks GA. Univariate analysis indicated that lower GA and birth weight were risk factors for intervention. There was a trend (*p* = 0.075, by Fisher’s exact test) suggesting that infants with both respiratory distress syndrome (RDS) and pneumothorax may be more likely (60%) to require intervention compared to infants with pneumothorax without RDS (29%). A larger pneumothorax ratio, measured in the anteroposterior (AP) (*p* < 0.0001) or lateral decubitus (DECUB) view (*p* < 0.008) was highly associated with need for intervention in this cohort of infants with pneumothorax. Of our 62 patients with radiographs, all had an AP view, but only 32 had a corresponding lateral decubitus film. Of the pneumothoraces analyzed, only 33% (n = 21) required intervention. Specifically, 1 neonate underwent needle thoracentesis (TC) alone, 7 received primary chest tube (CT) placement alone, and 13 initially underwent TC followed by CT placement due to persistent pneumothorax/clinical symptoms. Notably, 13 of the 14 neonates (93%) who received TC ultimately required subsequent CT placement.

In classifying outcomes descriptively, those patients with an anterior–posterior ratio (APr) of <0.102 (39%) required no intervention, while all of those with a high APr of >0.187 (15%) required intervention. For the MID ratio range 0.104–0.183, 47% required intervention. A Receiver Operator Characteristic (ROC) curve was generated for the anterior–posterior ratio for outcomes any intervention/no intervention. We chose the APr since it was always available (higher n) and the lateral decubitus was not always available, so it may not be as applicable in the clinical setting. The resulting area under the curve (AUC) was 0.902 (CI 0.828–0.977) and an APr of 0.184 was identified as the “best” cutoff (arrow, Figure 3). This cutoff value was associated with the highest Youden index (J statistic = 0.74) of any cutoff and had a sensitivity of 86%, a specificity of 88%, and correctly classified 87% of the subjects.

To further test the utility of the APr, we performed a multiple logistic regression analysis using a model that included the AP ratio, gestational age, use of CPAP in the delivery room (DR-CPAP), 5 min Apgar score, sex, and surfactant usage before the PTX was noted. With all variables present, the APr was still highly associated with intervention (*p*-0.002). After backward stepwise regression, both APr (*p* = 0.000) and DR-CPAP (*p* = 0.015) remained significantly independently associated with intervention. Of interest, DR-CPAP was protective. As can be seen in Table 1, of the PTX babies who did not require intervention, 83% (34/41) had been exposed to DR-CPAP, whereas for the PTX babies who did require intervention, significantly fewer, 38% (8/21), were exposed to DR-CPAP. Conversely, for the PTX babies with no DR-CPAP exposure, 13/20 (65%) ultimately got intervention for the PTX, while only 19% (8/42) of those exposed to DR-CPAP required intervention for their PTX.

## 4. Discussion

In this study, we evaluated whether the two-dimensional radiographic size of the pneumothorax, quantified as the ratio between the transverse measurement of the pneumothorax and the hemithorax size from either an AP or DECUB radiograph, could predict the need for procedural intervention in a cohort of primarily term neonates. Our results suggest that this measurement is a strong predictor of procedural intervention. Notably, none of the infants with a small pneumothorax ratio (APr < 0.102) required intervention, whereas all infants with a large pneumothorax ratio (APr > 0.187) underwent procedural management. These findings are consistent with our optimal APr cutoff from the AUC of 0.184 and correlate well with a prior study that noted pneumothorax size >20% or equivalent APr of 0.2 on initial chest x-ray was associated with need for chest drainage and worse prognosis for survival [11].

Our ROC curve analysis demonstrated high discriminative ability, with an AUC of 0.902, suggesting that the APr is a strong metric for predicting need for intervention. We suspect that the pneumothorax size ratio correlates strongly with the need for intervention because it serves as a proxy for severity. Larger pneumothoraces result in greater lung collapse, impaired gas exchange, and potentially adverse hemodynamic effects [12]. Approximately 40% of patients could be reliably classified as not needing intervention, and 15% as requiring it, based on either the AP or DECUB radiographic ratio alone (Figure 3). However, the predictive value of the “intermediate” range (APr 0.104–0.183) was uncertain. Further work is ongoing to explore whether combining a ratio in this mid-range with relevant clinical features, such as gestational age, respiratory distress syndrome (RDS), or clinical signs of respiratory decompensation, might improve its predictive accuracy, but more subjects are needed.

The use of CPAP in the delivery room (DR-CPAP) remained significant in the logistic regression analysis along with the AP ratio. However, it was a negative association; that is, the need for CPAP in the delivery room made it less likely that the baby would need intervention in a cohort of babies who all had a PTX. For the whole group (Table 1), the babies who did not get intervention were significantly more likely to have had DR-CPAP. We found this interesting since there are studies that suggest that CPAP can be associated with PTX [13,14]. However, it should be noted that our finding of DR-CPAP being protective for not needing intervention for a PTX is in a cohort of infants who all had PTX. This is a new finding which must be replicated in other cohorts for validity. In addition, our population is relatively small. However, we speculate that early CPAP may be stabilizing the alveoli in some way that makes it less likely to generate a larger size PTX that is more likely to require intervention. It may also be that even when a PTX is present, when the baby is on CPAP early on they are better supported and the baby does not have to take as deep a breath to support their own ventilation, which could exacerbate the PTX. We looked at the APr for the DR-CPAP babies and non-DR-CPAP babies and they were significantly different: median 0.119 (0.048, 0.204) vs. 0.232 (0.109, 0.384), *p* = 0.015 by Wilcoxon rank sum test. We also note both APr and no DR-CPAP were independently associated with intervention.

We also observed that smaller and lower gestational age infants were more likely to require intervention. Additionally, there was a signal to suggest that patients with a pneumothorax and RDS together were more likely to require intervention (29% RDS in intervention group vs. 10% in no intervention group, *p* = 0.075). This trend may be worth further investigation and with a larger sample size may become statistically significant as prior studies note an association between RDS and pneumothorax [15].

Surprisingly, nearly all infants who initially underwent needle thoracentesis (TC) ultimately required chest tube placement (13/14), suggesting limited utility for TC alone in this population. While it is possible that a larger sample size would improve accuracy for this finding, there is evidence in the literature that TC is often insufficient as a treatment alone. In one randomized control trial in neonates, they found that 55% of neonates initially assigned to needle aspiration subsequently required chest tube insertion within 6 h [16]. Two other studies cited the failure rate of needle aspiration, eventually requiring a chest tube at 58% and 37%, respectively [6,17]. In a recent 2018 Cochrane Review [18], the authors examined the effectiveness of thoracentesis. There were only two randomized trials [12,19] that could be included, with a total of 142 infants enrolled. An immediate CT strategy did not offer any advantages, while performing the TC with an angiocatheter that remains in situ post-procedure may offer less need for later CT. Overall, about 30% of the babies treated with TC did not go on to need CT. In the adult literature, failure rates of needle aspiration were 83% and 82%, respectively [20,21]. Clearly, there are not many studies on which we can rely. In addition, since our study was a retrospective study, we cannot ascertain with certainty whether some of the needle aspirations were done immediately before the chest tube as a temporizing measure with a chest tube already planned. Additionally, we can speculate that compared to historical practice patterns, we are intervening less with needle aspiration on smaller PTXs than in the past.

These findings have practical and clinical implications. Early identification of neonates at higher or lower risk for needing procedural intervention may help optimize staffing and preparedness in neonatal intensive care units (NICUs), particularly when specialized personnel are required for TC or CT placement.

This study has several limitations. First, although we searched the old EMR with the same ICD and CPT codes, some earlier charts may not have been coded consistently and thus may not have been retrieved. This raises the possibility of limited under-ascertainment prior to the EMR transition in 2023. Additionally, the relatively small sample size, driven in part by recent changes to our EMR system, may have affected the statistical power and precision of some of our findings. Second, although the radiographic pneumothorax ratio appears predictive, clinical decisions are multifactorial and include assessment of the neonate’s respiratory and hemodynamic status, and these clinical features were not controlled for in our study. Third, our data were derived retrospectively from a single center, raising potential issues for external validity. This also raises the issue of local practices and indications for TC or CT in the setting of a pneumothorax and clinical respiratory distress. We do not have a protocol to guide this clinical decision. Finally, the lack of inter-rater reliability due to a single measurer strengthens the internal validity of our study, but could represent a limitation regarding the reproducibility of the method across different clinicians.

In conclusion, radiographic pneumothorax size appears to be a useful tool in predicting the need for intervention in neonates with pneumothorax, particularly at the extremes of size. Future research should aim to validate these findings in larger cohorts, including at other centers, with more lower GA infants, and assess whether incorporating clinical variables can improve prediction in cases with intermediate pneumothorax size ratio.

## Figures and Tables

**Figure 1 children-13-00041-f001:**
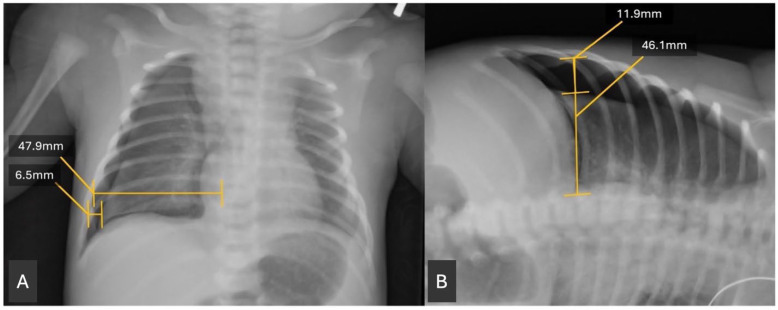
(**A**,**B**): Example of pneumothorax (PTX) measurements. Transverse lines at the widest part of the PTX and the widest part of that hemidiaphragm are drawn and measured on both an AP film (left panel) and similarly on a lateral decubitus film if available (right panel). A simple ratio is created of the PTX/hemidiaphragm measurements. For A, ratio is calculated as follows: 6.5/47.9 = 0.136. For B, ratio is calculated as follows: 11.9/46.1 = 0.258.

**Figure 2 children-13-00041-f002:**
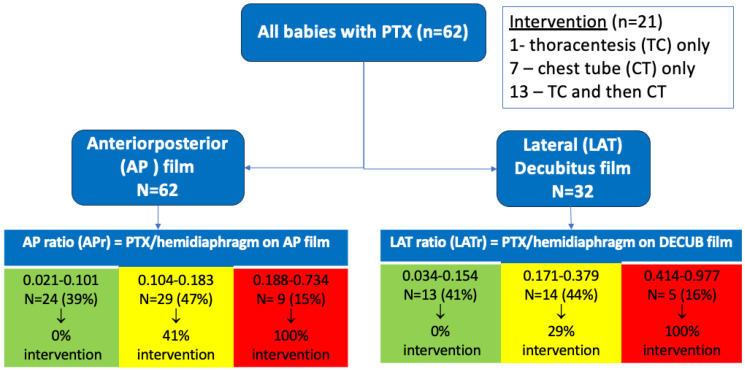
The patient flow diagram shows the outcome of intervention (thoracentesis and/or chest tube) vs. no intervention and the resulting radiograph-derived ratios categorized by outcome.

**Figure 3 children-13-00041-f003:**
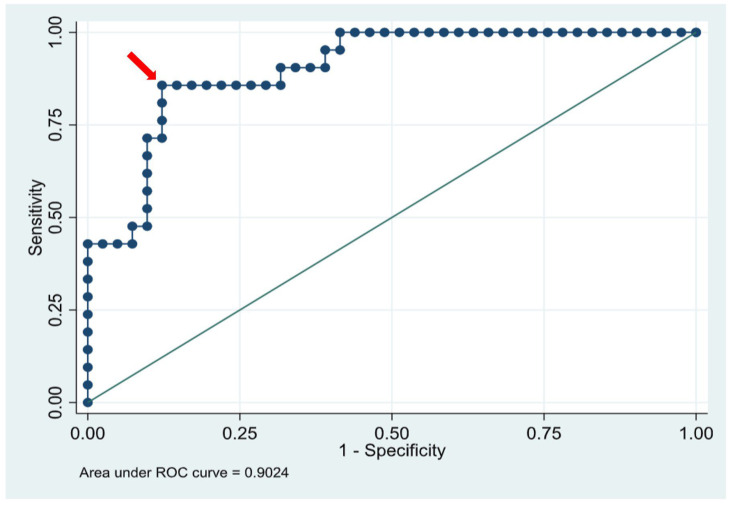
Receiver Operator Characteristic (ROC) curve was generated for the AP ratio (APr) as a continuous variable for the outcome any intervention (thoracentesis and/or chest tube) vs./no intervention. AUC = 0.902 (CI 0.828–0.977), *n*—62 babies with pneumothorax, 21 received intervention. The red arrow denotes the optimal ratio cutoff of 0.184.

**Table 1 children-13-00041-t001:** Patient Demographics.

	All (n = 62)	Intervention ^ (n = 21)	No Intervention (n = 41)	*p* Value
Birth weight (grams)	3028.5 (2600, 3630)	2690 (980, 3160)	3345 (2750, 3710)	0.0063 *
Gestational Age (weeks)	38.15 (36.1, 39.3)	36.3 (26.3, 38.3)	38.6 (37, 39.6)	0.0054 *
Sex (% male)	42 (68)	15 (71)	27 (66)	0.027
C-section delivery	37 (60)	15 (71)	22 (54)	0.274
Required CPAP in Delivery Room	42 (68)	8 (38)	34 (83)	0.001 *
Required PPV in Delivery Room	18 (29)	9 (43)	9 (22)	0.086
Surfactant Given Prior to PTX Identification	11 (18)	9 (43)	2 (5)	0.001 *
Surfactant Given During Admission	20 (32)	14 (67)	6 (15)	<0.0001 *
Diagnosis of RDS	10 (16)	6 (29)	4 (10)	0.075
Anteriorposterior (AP) Ratio on Radiograph	0.14 (0.7, 0.25)	0.27 (0.21, 0.47)	0.08 (0.04, 0.14)	<0.0001 *
Lateral Decubitus (DECUB) Ratio on Radiograph	0.1 (0.10, 0.30)	0.41 (0.26, 0.52)	0.15 (0.09, 0.21)	0.0003 *
Respiratory Rate at time of PTX	55.5 (44, 79)	60 (49, 84)	53 (43,72)	0.2056
Oxygen saturation at time of PTX	96 (93, 99)	92 (85, 93)	97 (95, 99)	<0.0001 *
Respiratory Distress on Exam	49 (79)	18 (86)	31 (76)	0.355
Respiratory support at PTX Diagnosis				<0.0001 *
Room Air	11 (18)	0 (0)	11 (27)	
NC ≤ 2 L	6 (10)	3 (14)	3 (7)	
CPAP	37 (60)	11 (52)	26 (63)	
Mechanical Ventilation	8 (13)	7 (33)	1 (2)	
Intubated before PTX development	12 (19)	10 (48)	2 (5)	<0.0001 *
Intubated prior to PTX resolution	24 (39)	17 (85)	7 (17)	0.000 *
Days of Mechanical Ventilation	0 (0, 3)	3 (1, 7)	0 (0, 0)	0.0000 *

Data are presented as median (IQR), or n (%). ^ intervention as thoracentesis and/or chest tube. * marks statistical significance between the two groups. CPAP, Continuous Positive Airway Pressure; PPV, Positive Pressure Ventilation; PTX, Pneumothorax; RDS, Respiratory Distress Syndrome; NC, Nasal Cannula; all continuous variables were not normally distributed and *p* values reflect Wilcoxon rank sum testing; the categorical variables were tested using chi-square or Fisher’s exact test.

## Data Availability

The data presented in this study are available on request from the corresponding authors. The data are not publicly available due to privacy and ethical reasons.

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
