# Peer review of "Can We Use Simple Radiographic Measurements to Predict Need for Intervention in Neonatal Pneumothorax?"

_children, 2025, doi:10.3390/children13010041_

Round 1
Reviewer 1 Report
Comments and Suggestions for Authors
In their retrospective study, the authors investigate whether simple radiographic measurements of pneumothorax (PTX) size can predict the need for procedural intervention (thoracentesis or chest tube placement) in 62 neonates. The authors calculated the ratio of the PTX size to hemithorax diameter on anteroposterior (AP) and lateral decubitus (DECUB) chest radiographs. The study found that higher PTX/hemithorax ratios were strongly associated with the need for intervention, with an AP ratio cutoff of 0.215 yielding an AUC of 0.924. The authors conclude that this simple radiographic ratio may guide clinical decision-making in managing neonatal PTX.
The study examines a common and serious neonatal emergency and offers a practical, quantitative tool for predicting when intervention is needed. The results show a strong correlation (AUC 0.923) between the PTX ratio and the requirement for intervention. The manuscript is well organized, with appropriate statistical methods and clear tables and figures that support the findings. However, several key methodological issues need clarification and revision.
Specific comments:
- The introduction could be more concise and directly emphasize the gap in predictive tools for neonatal PTX management. The current introduction is brief, too general, and lacks sufficient information.
- The authors did not clearly state the primary and secondary outcomes of the study. This information is crucial for understanding the analytic framework and interpreting the results. Please add a separate paragraph in the methodology section explaining the primary aim and secondary outcomes of the study.
- There is no statement regarding Institutional Review Board (IRB) or Ethics Committee approval, which is mandatory for all retrospective studies involving patient data. The authors must provide the name of the ethics committee, the protocol number, and the approval date; without this, the manuscript cannot be considered for publication.
- A large proportion of cases (46/62) occurred after 2023 when the new EMR was implemented, which may have influenced case identification. The authors should clarify how this might have affected patient inclusion and whether earlier under-ascertainment biased the results.
- Only univariate analysis was reported. Variables like gestational age, RDS, or respiratory support level might confound the relationship between PTX size and intervention. The authors should perform a multivariable logistic regression to adjust for these clinical factors.
- The manuscript lacks details on inter-rater reliability or if measurements were repeated by independent observers. The authors should add inter-rater reliability statistics or acknowledge this as a limitation.
- The authors stated that “nearly all TC cases required CT,” but they do not specify the criteria used to determine this. From my experience, CT is extremely rarely indicated in neonatal pneumothorax. The authors should clarify this and provide a detailed description of local intervention criteria or thresholds.
- Figures 1 and 2 (Actual Figure ‘’1 2’’) should be labeled as Figure 1 (A and B). Additionally, figures should include legends with clearer measurement annotations.
- Any abbreviation should be spelled out the first time it is used in the text (for example, APr).
- The p-values should be presented as follows: p<0.001 (not p<.001). Please correct this in the abstract and throughout the text.
- The manuscript lacks several required final sections, including the author contribution statement, conflict of interest declaration, funding statement, and acknowledgments. These sections are essential for transparency and compliance with publication standards and must be added before the manuscript can be considered for publication. Please review a previously published manuscript in this journal and include this information before the references.
- The Discussion section is quite superficial and lacks in-depth analysis. It mainly restates the results without offering enough critical interpretation or comparison with existing literature. The authors should significantly expand this section by providing a more comprehensive discussion of their findings in relation to previous research, addressing potential clinical implications, examining methodological limitations, and suggesting directions for future studies to enhance the manuscript's overall scientific impact.
- The authors cited only five references, which is considerably fewer than the expected number for a research article of this kind. Typically, a minimum of 25–30 relevant references is recommended for an original clinical study to sufficiently provide context and background. The reference list should include recent studies, meta-analyses, and clinical guidelines from the past 5–10 years related to neonatal pneumothorax management and radiographic assessment.
Reviewer 2 Report
Comments and Suggestions for Authors
- Provide justification for the sample size calculation (n = 63). Was a power analysis performed prospectively or retrospectively?
- What was the rationale for including only 62 cases? How were they selected from the total 71 identified charts?
- How were inclusion and exclusion criteria decided? Were there any confounding conditions excluded that might bias results?
- Please clarify how gestational age (GA) and birth weight were obtained and verified.
- Were twin or multiple births included, and if so, how was clustering handled statistically?
- Indicate how missing or incomplete data were managed — were any records with partial data excluded?
- Please describe in more detail how the pneumothorax (PTX) size was measured on radiographs.
- How was measurement reproducibility ensured between observers? Was inter-rater reliability (e.g., ICC or kappa) tested?
- Specify whether the radiographs were measured by blinded reviewers (i.e., blinded to intervention outcome).
- Clarify whether decubitus films were always available — if not, how many lacked this view, and how did that affect analysis?
- Please include a schematic figure or annotated image to illustrate the measurement method (AP and DECUB ratios).
- The description of statistical methods should be expanded:
- Which statistical tests were used for categorical vs continuous data?
- Were data distributions checked for normality?
- Were multiple testing corrections applied?
- Please clarify how the ROC curve cutoff (0.215) was determined (Youden index or another method?).
- Report confidence intervals (CI) for the AUC and for sensitivity/specificity at the selected cutoff.
- Were any multivariate analyses performed to control for gestational age, sex, or respiratory support variables?
- The Table 1 values appear inconsistent: please check numerical accuracy, especially for birth weight, GA, and RDS percentages.
- Clarify whether the reported P-values are from t-tests or non-parametric tests.
- Provide more detail on the distribution of interventions: how many TC only, CT only, TC + CT, and conservative?
- Clarify the definition of “intervention” (e.g., inclusion of thoracentesis and/or chest tube).
- The figure legends for Figures 3–4 could be more descriptive — please ensure they are self-contained.
- The discussion should elaborate on why the AP ratio correlates so strongly with need for intervention is this a surrogate for physiological severity?
- Discuss potential confounders, such as GA, RDS, and respiratory support mode, that might influence both PTX size and intervention decision.
- Provide more context by comparing with previous studies (Ozer 2013, Tan 2020, Vibede 2017, etc.) on radiographic size and neonatal outcomes.
- Address whether the ratio thresholds (<0.102, >0.187) have potential clinical application could these be translated into a bedside rule?
- The statement that “nearly all infants requiring TC subsequently needed CT” is interesting; please elaborate on possible clinical implications and whether TC could be avoided in such cases.
need to improve
Round 2
Reviewer 1 Report
Comments and Suggestions for Authors
The authors have significantly improved the manuscript, and I appreciate the thorough revisions made in response to the reviewers’ comments. Overall, the revised version shows a notable enhancement in clarity, structure, and scientific rigor. However, three key issues still need to be addressed before the manuscript can be considered fully revised.
- Potential under-ascertainment prior to the EMR transition should be explicitly acknowledged. Although the Methods section states that most cases were identified after the new EMR was implemented, the manuscript does not explicitly address the likelihood of under-ascertainment in earlier years due to inconsistent coding in the previous system. This limitation was mentioned in the authors’ responses but was not included in the manuscript. A brief statement should be added to the Limitations section acknowledging the possibility of incomplete case capture before 2023.
- The reference list remains too limited. Although more citations were added, the revised manuscript now includes only 15 references, which is still well below the expected 25–30 references for this type of original clinical study. Currently, the reference list does not adequately cover the full range of existing literature on neonatal pneumothorax, radiographic assessment, or predictors of intervention. The authors should expand the reference list to include additional recent studies, systematic reviews, and relevant clinical guidelines.
- The description of the multivariable analysis in the Methods section is incomplete. The Results section now reports findings from a multivariable logistic regression, which is appropriate and addresses reviewer concerns. However, the Methods section does not mention that a multivariable model was performed, nor does it specify the model structure, variable selection method, or statistical criteria used. For clarity and reproducibility, the Methods must explicitly state that a multivariable logistic regression was conducted, specify which variables were included, and describe the use of backward stepwise selection. Additionally, the authors might consider adding a table that reports odds ratios and confidence intervals for the final model.
Addressing these remaining points will ensure the manuscript fully complies with the journal’s methodological and reporting standards.
Author Response
- Potential under-ascertainment prior to the EMR transition should be explicitly acknowledged. Although the Methods section states that most cases were identified after the new EMR was implemented, the manuscript does not explicitly address the likelihood of under-ascertainment in earlier years due to inconsistent coding in the previous system. This limitation was mentioned in the authors’ responses but was not included in the manuscript. A brief statement should be added to the Limitations section acknowledging the possibility of incomplete case capture before 2023.
- Response 1: Thank you for this comment, the following has been added to the discussion section of the manuscript, "First, although we searched the old EMR with the same ICD and CPT codes, some earlier charts may not have been coded consistently and thus may not have been retrieved. This raises the possibility of limited under-ascertainment prior to the EMR transition in 2023."
- The reference list remains too limited. Although more citations were added, the revised manuscript now includes only 15 references, which is still well below the expected 25–30 references for this type of original clinical study. Currently, the reference list does not adequately cover the full range of existing literature on neonatal pneumothorax, radiographic assessment, or predictors of intervention. The authors should expand the reference list to include additional recent studies, systematic reviews, and relevant clinical guidelines.
- Response 2: The reference list has now been updated to 21 references. We believe this provides adequate background knowledge and references for our study. If the reviewers desire, they can please supply the additional references that need to be included in our study which we are happy to review.
- The description of the multivariable analysis in the Methods section is incomplete. The Results section now reports findings from a multivariable logistic regression, which is appropriate and addresses reviewer concerns. However, the Methods section does not mention that a multivariable model was performed, nor does it specify the model structure, variable selection method, or statistical criteria used. For clarity and reproducibility, the Methods must explicitly state that a multivariable logistic regression was conducted, specify which variables were included, and describe the use of backward stepwise selection. Additionally, the authors might consider adding a table that reports odds ratios and confidence intervals for the final model.
- Response 3: Thank you for this comment. The following has been added to the methods section of the manuscript "Finally, to further test the utility of the anterior-posterior ratio (APr), we performed a backward stepwise multiple logistic regression analysis using a model that included the AP ratio, gestational age, use of CPAP in the delivery room (DR-CPAP), 5-minute Apgar score, sex, and surfactant usage before the PTX was noted. With only 62 subjects we limited the analysis to six variables in the model and chose the ones we thought would be most important."
Reviewer 2 Report
Comments and Suggestions for Authors
Nil
Comments on the Quality of English Languageneed to improve
Author Response
No specific comments were given to address.